# A Complete Automatic Target Recognition System of Low Altitude, Small RCS and Slow Speed (LSS) Targets Based on Multi-Dimensional Feature Fusion

**DOI:** 10.3390/s19225048

**Published:** 2019-11-19

**Authors:** Qi Wu, Jie Chen, Yue Lu, Yue Zhang

**Affiliations:** National Key Laboratory of Science and Technology on Automatic Target Recognition, National University of Defense Technology, Changsha 410073, China; wuqi13@nudt.edu.cn (Q.W.); chenjie13@nudt.edu.cn (J.C.); bit_drinkwater@163.com (Y.L.)

**Keywords:** low altitude, small RCS and slow speed targets, automatic target recognition, multi-dimensional feature extraction, multi-layer classifier system, directed acyclic graph, micro-Doppler, holographic staring radar

## Abstract

Low altitude, small radar cross-section (RCS), and slow speed (LSS) targets, for example small unmanned aerial vehicles (UAVs), have become increasingly significant. In this paper, we propose a new automatic target recognition (ATR) system and a complete ATR chain based on multi-dimensional features and multi-layer classifier system using L-band holographic staring radar. We consider all steps of the processing required to make a classification decision out of the raw radar data, mainly including preprocessing for the raw measured Doppler data including regularization and main frequency alignment, selection, and extraction of effective features in three dimensions of RCS, micro-Doppler, and motion, and multi-layer classifier system design. We design creatively a multi-layer classifier system based on directed acyclic graph. Helicopters, small fixed-wing, and rotary-wing UAVs, as well as birds are considered for classification, and the measured data collected by L-band radar demonstrates the effectiveness of the proposed complete ATR classification system. The results show that the ATR classification system based on multi-dimensional features and k-nearest neighbors (KNN) classifier is the best, compared with support vector machine (SVM) and back propagation (BP) neural networks, providing the capability of correct classification with a probability of around 97.62%.

## 1. Introduction

Recently, with the rapid development of small unmanned aerial vehicles (UAVs) technology, all kinds of low altitude, small radar cross-section (RCS) and slow speed (LSS) aircrafts are increasing in number, and often flying illegally, even being used in terrorist attacks [1,2]. LSS targets such as small UAVs have become a serious air threat in both military and civil aspects [3]. Effective early warning, detection, tracking, and identification of LSS targets have become an increasingly important task. Effective detection of LSS targets mainly relies on radar, which can work all-day and in all weather conditions and obtain important parameters such as range, velocity, and category attributes of all kinds of targets [4]. However, due to their small radar cross-section (RCS), their echo signals are extremely weak. Their flight heights are low and their velocities are slow, which make the echoes drown in a background of clutter [2]. Moreover, flying birds are the main obstacles for detecting LSS targets, leading to many false alarms. Due to the above factors, conventional mechanical scanning radar and phased array radar have many detection problems such as low Doppler resolution, low data rate, high false alarm rate, and weak classification ability, which results in difficulty meeting the requirements of early warning and detection for LSS targets such as UAVs.

Holographic staring radar (also known as full time-space radar, ubiquitous radar) is an effective means to realize the detection and recognition of LSS targets [5]. It originated from the concept of ubiquitous radar [6] proposed by professor Skolnik, a famous British radar expert, around 2000. The core idea is to adopt a digitized array for receiving and transmitting, a wide beam for transmitting, and a digital multi-beam for receiving to cover the transmitting area, so as to realize continuous detection of time domain and airspace for the whole observed airspace [7]. Due to the advantages of the full time-space coverage and 100% target dwell time of holographic staring radar, long-term phase-coherent integration can obtain sufficient processing gain and fine Doppler resolution, so as to realize effective separation of weak and small targets from clutter, and provide a new micro-Doppler dimension for target classification and recognition.

Over the years, micro-Doppler signatures that represent the micro-motion of the parts of the target have been widely utilized for small UAV recognition [8,9]. In [10], the authors analyzed in detail the micro-Doppler effect of sinusoidal modulation caused by vibration and rotation in micro-motion, and simulated the micro-Doppler frequency shifts in the frequency spectrum. In [11], they proposed that phase modulation induced by a target’s micro-motion caused micro-Doppler signatures. The unique micro-Doppler information and relevant pseudo-Zernike moments extracted from the cadence velocity diagram (CVD) can describe the characteristics of the target’s micro-motion, helping the characterization and classification of targets. In [5], the Aveillant company successfully identified drones and birds based on the rotary-wing UAV’s unique Doppler plot, using Gamekeeper 16U phased array radar. The researchers in [12] used a convolutional neural network (CNN) to train and classify the micro-Doppler time-frequency diagram and cadence velocity diagram (CVD) of UAVs, and their accuracy reached 94.7%. In [13], the authors proposed that the Global Navigation Satellite System (GNSS) can be used as an illuminator for a passive radar system and focused on the case of helicopter rotor blades where the Doppler shifts are very high, demonstrating the possibility of detecting the kind of targets and measuring their micro-Doppler signatures. In [14], they introduced the extraction of micro-Doppler features of helicopter and human targets using the wavelet transform method incorporated with time-frequency analysis. Analysis on micro-Doppler features can estimate the target’s micro-motion parameters, such as the vibration/rotation rate, and be successfully applied to both helicopter and human data. In [15,16], Rahman and Robertson used coherent W-band radar to apply a wavelet transform to radar data collected on a DJI phantom and bionic birds. The experimental results show that the improved Doppler resolution provides more useful information, which is conducive to the classification of small UAVs. In [15,17], the QinetiQ company used X-band FMCW radar to explore specific Doppler patterns of various UAVs during flight, proving that the unique Doppler plot of the target is an important prospect for target identification in future studies. In [18], the authors incorporated different micro-Doppler-based classification techniques to classify warheads and confusing objects.

In this paper, the influence of the number of rotary-wings on the amplitude and phase components of the radar echo is studied first. Then features of three dimensions of micro-Doppler, RCS, and motion are analyzed and selected based on the simulated data and real measured data in the out-field. Then a complete automatic target recognition (ATR) system of LSS targets based on multi-dimensional feature fusion is proposed. The successive steps in the ATR chain are: The generation of a Doppler modulation spectrum (DMS) obtained by processing pulse compression and moving target detection (MTD) long-time integration; preprocessing for raw measured DMS including strong noise reduction near zero frequency, regularization for DMS to enlarge the weak micro-Doppler frequency components, and main frequency alignment to compensate the main Doppler shift caused by the target’s body motion; multi-dimensional feature extraction in three diminsion of RCS, micro-Doppler and motion; multi-layer classifier system design; and finally, classification of various LSS targets. Experimental results based on the measured data show that the directed acyclic graph classification system based on multi-dimensional feature fusion and the K-nearest neighbors (KNN) classifier is the best for the several typical LSS targets such as helicopters, small fixed-wing and rotary-wing UAVs, as well as birds. The classification result of the multi-dimensional feature fusion classification system is better than that of the equal-weighted classification system with multiple single-dimension feature classifiers, and the KNN classifier is better than the support vector machine (SVM) classifier and the back propagation (BP) neural network classifier. Experimental results show that for several LSS targets, the proposed best classification system can obtain the capability of correct classification with around 97.62%.

The main contributions of this study can be summarized as follows:A complete automatic target recognition (ATR) classification system of several typical LSS targets based on multi-dimensional features is developed;The successive steps of the complete ATR chain are discussed based on real measured radar data;The influence of the number of rotary-wings on the amplitude and phase components of radar echo in the time domain and frequency domain are modelled and simulated;A multi-layer classifier system design based on directed acyclic graph is proposed.

The rest of the paper is organized as follows. The influence of the number of rotary-wings on the radar echo is given in Section 2. The micro-Doppler signatures analysis and RCS and motion features analysis are discussed in Section 3. The experimental setup, preprocessing for measured DMS and real measured radar data are introduced in Section 4. The multi-layer classifier system design and classification results are analyzed and concluded in Section 5. Finally, conclusions are drawn in Section 6.

## 2. The Influence of the Number of Rotary-Wings

Many aircrafts have rotating parts, such as helicopters and rotary-wing UAVs. The micro-motion of rotary-wings relative to the fuselage causes the additional frequency modulation on the radar echo signal, producing additional micro-Doppler side frequencies around the main Doppler shift induced by the main body of the target, known as micro-Doppler effects [19,20,21,22,23,24,25]. Usually, helicopters have a main rotary-wing while rotary-wing UAVs have four, six, and other rotary-wings, called quadcopter, hexacopter, etc. The influence of different numbers of rotary-wings on the radar echo in the time domain and frequency domain are simulated and analyzed.

First, we analyze the expressions of modulation signals in the time domain and frequency domain generated by single rotary-wing and quad rotary-wings in the far field, respectively. A single rotary-wing consists of a rotor and N blades with blade length as l, which are homogeneous rigid bodies that move with the target’s body and rotate around the rotor. Quad rotary-wings consist of four of the same rotary-wings, evenly distributed in the four corners of the fuselage. We suppose that the lengths from the four rotary-wings to the central point Q are the same as L, and their initial angles are shown as $\psi_{k}=\psi_{0}+2\pi k/M,k=0,1,2,3$, where M=4 is the number of rotary-wings. Every rotary-wing has N blades with blade lengths as l, and N=2 is the number of blades in a rotary-wing.

To simplify the model, the whole target is regarded as multiple scattering points. The geometry of the single and quad rotary-wings and the radar are shown in Figure 1 and Figure 2, where the radar is stationary and located at the origin O of the radar coordinate system (X,Y,Z), the rotation center of the target is located at the origin Q of the target coordinate system (x,y,z). Both of the single and quad rotary-wings move with velocity v and radial angle φ between the direction of target flight and the radar. Their blades rotate with rotating angular velocity $w_{r}$ (rotational frequency $f_{r}$, $w_{r}=2\pi f_{r}$) on the plane (x,y,z=0) around their respective rotor. The azimuth angle and elevation angle of the origin Q in the radar coordinate system (X,Y,Z) are α, and β, respectively, and the initial distance from the origin Q to the origin O is $R_{0}$. For the scattering point P io one of the blades, its distance to the origin Q is $l_{p}$ and its initial angle is $\theta_{0}$. Then, at time t, for the single rotary-wing M=1 and quad rotary-wings M=4, their time domain radar echo signals synthesized by M rotary-wings and N blades in each rotary-wing are expressed as follows in Equation (1) [26]. For quad rotary-wings M=4, two symmetrical rotors rotate clockwise, to balance the moment, and the other two symmetrical rotors rotate counterclockwise, shown as $w_{r}$ and $-w_{r}$. We assume that the carrier frequency $f_{c}$ and the main Doppler frequency $f_{d}$ caused by the target’s translational motion have been compensated ($f_{d}=2v\cos\varphi /\lambda$).
(1)$$\begin{array}{ll} s_{N}(t) & =\sum_{K=0}^{M-1}\sum_{k=0}^{N-1}s_{l}(t)\\ & =\sum_{k\prime =0}^{M-1}\sum_{k=0}^{N-1}l\sin c\{\frac{2\pi }{\lambda }l\cos\beta \cos\left(\theta_{0}+\frac{2\pi k}{N}+{\left(-1\right)}^{k\prime }w_{r}t-\alpha \right)\}\times \\ & \exp\{-j\frac{4\pi }{\lambda }[R_{0}+\frac{l}{2}\cos\beta \cos\left(\theta_{0}+\frac{2\pi k}{N}+{\left(-1\right)}^{k\prime }w_{r}t-\alpha \right)]\} \end{array}$$

From Equation (1), it can be seen that the slight movement of any part on the rotary-wings will cause fluctuations of amplitude and phase components of radar modulation echo, and the rotating motion of blades on the rotary-wings at an angular velocity $w_{r}$ will have a periodic modulation effect on the amplitude and phase components of the base-band modulated echo.

Now, we analyze the expressions of modulation signals in the frequency domain generated by single rotary-wing and quad rotary-wings in the far field, respectively. According to the property of the Bessel function [27]:
(2)$$e^{jz\sin\theta }=\sum_{n=-\infty }^{+\infty }J_{n}(z)e^{jn\theta }$$

We transform Equation (1) using the Bessel function Equation (2) and then carry out the Fourier transform (FT), transforming the expression (1) of modulation signals in time domain into the expressions (3) and (4) in the frequency domain, as follows [28]:(3)$$S_{N}(f)=\sum_{K=-\infty }^{+\infty }2\pi C_{K}\delta (f-KNf_{r})$$
(4)$$\begin{array}{l} C_{K}=N\exp(jKN\theta_{0})\\ \sum_{i=0}^{M-1}\exp\{j\frac{4\pi }{\lambda }(R_{0}+L\cos\beta \cos\left(\psi_{0}+2\pi i/M-\alpha \right))\}\int_{0}^{l}J_{KN}(\frac{4\pi }{\lambda }l_{p}\cos\beta )dl\\ \approx MN\exp(jKN\theta_{0})\exp\{j\frac{4\pi }{\lambda }R_{0}\}\int_{0}^{l}J_{KN}(\frac{4\pi }{\lambda }l_{p}\cos\beta )dl \end{array}$$

In which, the value of $C_{K}$ is shown in Equation (4). For the single rotary-wing, M=1 and L=0, while for the quad rotary-wings, M=4 and L≠0. In which, $\delta (f-KNf_{r})$ is called the Dirac delta function, a generalized function that describes the distribution density of points, and it is defined into $\delta (f-KNf_{r})=\{\begin{array}{cc} \infty , & f=KNf_{r}\\ 0 & f\neq KNf_{r} \end{array}$. Equations (3) and (4) indicate that the modulation spectrums of single rotary-wing and quad rotary-wings both consist of a series of spectral lines, with the interval $Nf_{r}$ between spectral lines. The interval is only decided by the number N of blades in each rotary-wing and their rotational frequency $f_{r}$. Their difference lies only in the value of $C_{K}$. The $C_{K}$ of quad rotary-wings is four times larger than that of the single rotary-wing, both related to the number of rotary-wings M, the number of blades in each rotary-wing N, the wavelength of radar λ, the initial angle $\theta_{0}$, the length of blades l, the elevation angle β, and the Bessel function.

According to the above analysis, the following conclusion can be drawn: For the rotary-wings with the same blade parameters, the number of rotary-wings does not affect the frequency distribution of the modulated echo in the frequency domain, but only affects the amplitude of each frequency component, that is, the number of rotary-wings only affects the amplitude component of the radar echo signal, not the phase component.

In the following, we will simulate the radar echoes of single rotary-wing and quad rotary-wings. In the simulation experiments, assume the carrier frequency of radar $f_{c}=1.3$ GHz, pulse repetition frequency PRF=5 kHz, radar bandwidth is 2 MHz, the initial distance of the target and radar $R_{0}=40$ km in far field. Single and quad rotary-wings both have two blades in every rotary-wing with blade length l=0.1 m, rotational frequency $f_{r}=80$ Hz, moving velocity v=10 m/s, and SNR=20 dB. According to the above conditions and parameters, radar echo signals in time domain and frequency domain of single rotary-wing and quad rotary-wings can be shown as follows, respectively:

From Figure 3 and Figure 4, the time domain modulation echoes of both single rotary-wing and quad rotary-wings are a set of pulse train with a modulation interval of Δt=6.4 ms, corresponding to sinc function in echo Equation (1). Both the Doppler domain modulation spectrums consist of a series of spectral lines with the main frequency $f_{d}$ = −87.89 Hz, caused by the main velocity of the target’s translational motion, and multiple side frequencies symmetrically distributed with their interval $f_{T}=159.93$ Hz. The interval of the spectrum lines is also called periodic modulation frequency as $Nf_{r}$, only determined by the number N=2 of blades in each rotary-wing and its rotational frequency $f_{r}$ = 80 Hz, corresponding to Equation (3).

The above simulation results and analysis indicate that frequency distributions of both the Doppler domain modulation spectrums are the same, however, the amplitude of each frequency component of quad rotary-wings is close to four times than that of single rotary-wing. The simulation results are consistent with the theoretical analysis, verifying that the number of rotary-wings only affects the amplitude component of the radar echo signal, not the phase component, in the far field.

## 3. Multi-Dimensional Feature Analysis

### 3.1. Micro-Doppler Features

Micro-Doppler information is caused by the modulation effect of micro-motion of the moving parts of the target on the radar echo [18,29]. It represents the subtle structural and motion information of the moving parts of the target. It is the unique information that can distinguish different targets. The Doppler modulation spectrum (DMS) of the target represents the Doppler frequency distribution and the amplitude of each frequency component of the target’s echo signal. Frequency components of the DMS consist of the main frequency and micro-Doppler side frequencies [17]. The main frequency component is the main Doppler frequency shift induced by the main motion of the target’s body. The micro-Doppler side frequency components located on both sides of the main Doppler frequency are caused by micro-motion of the moving parts of the target, such as the spinning of propeller blades or the flapping motion of a bird’s wings. Micro-Doppler features can be extracted from DMS, such as the number of side frequencies, periodicity of side frequencies (i.e., the interval between them), symmetry of side frequencies, the maximum micro-Doppler side frequency, etc.

In the actual environment, due to the weak target signal being often submerged in the clutter, in order to solve the detection difficulty of LSS targets, a long-term multi-pulse integration algorithm is often adopted to effectively improve the signal to noise ratio (SNR) gain of echo, so as to improve the detection ability of radar for weak targets [30,31]. The holographic staring radar system studied in this paper can continuously dwell on targets over the entire search volume for a long time. Utilizing pulse compression and MTD long-term integration algorithm for the measured radar echo data can achieve a sufficient processing gain and fine DMS with a high Doppler resolution [7]. The specific process is shown in Figure 5 below:

According to the above steps mentioned, we simulate the DMS of typical LSS targets and then compare the simulated DMS with the measured DMS. Taking quadcopter, helicopter, and bird as examples, their respective radar modulated echo signals are first constructed, and then the echo signals are processed by pulse compression and long-term integration of MTD to obtain their respective simulated range-Doppler (RD) plane and simulated single-frame DMS. In the simulation, the radar and target parameters are shown in Table 1. The initial distance between the target and the radar is 2 km, and flying height is 100 m. We take 400 range units and accumulate 8192 pulses, to obtain the simulated RD plane and DMS of quadcopter, helicopter, and bird, as shown in Figure 6, Figure 7 and Figure 8, respectively:

From the simulated RD planes of the three typical LSS targets, we can see that in the target echo signal after dealing with the accumulation of MTD on the range axis, target peaks account for multiple range units. This is because the movement of the target may cause the echo signal between multiple pulses to cross the range units, which is called range migration. Therefore, it is necessary to optimize the traditional MTD accumulation algorithm to compensate for range migration. However, this phenomenon does not affect DMS with high resolution of the target, representing the Doppler frequency distribution and the amplitude of each frequency component of the target’s echo signal.

From the comparison results of simulated and real measured DMS of three typical LSS targets, whether for simulation or the real measurement, three of the DMS are composed of the main frequency component and the micro-Doppler side frequencies. The spectral line with the largest amplitude is the main frequency component, which is the Doppler shift generated by the body movement of the target. The side frequencies distributed symmetrically on both sides of the main frequency are caused by the micro-motion of the moving parts, such as the spinning of the propeller blades of the quadcopter and helicopter, and the flapping motion of a bird’s wings, and their amplitudes are lower by dozens of dB, compared to the main frequency.

In the DMS of the quadcopter, the number of spectral lines is relatively less and sparse, the interval of spectral lines (i.e., periodic modulation frequency) is relatively large, and the spectral width is relatively wide. In the DMS of helicopter, the number of spectral lines is more and denser, the interval of spectral lines (i.e., periodic modulation frequency) is relatively small, and the spectral width is extremely wide. In the DMS of the bird, the spectral lines are basically concentrated in the main frequency, and several dense spectral lines are distributed on both sides of the main frequency, with extremely narrow spectral widths. The comparison results are shown in Table 2:

From the above table, the three typical LSS targets of quadcopter, helicopter, and bird have three obvious micro-Doppler informative features, including spectral line number, periodic modulation frequency, and spectral width.

### 3.2. RCS Features

Radar cross-section (RCS) describes the scattering ability of a radar target to electromagnetic waves hitting the target’s surface. The target’s RCS changes from up and down with the posture angle. As though the posture angle of a complex target (containing multiple scattering points) has a small change, its RCS value will have a large up and down fluctuation. The power modulation of the LSS target’s movement on radar echo is mainly reflected on the target’s RCS sequence over a period of time. Therefore, the posture and motion information of the target can be extracted from the change rule of RCS sequence. In addition, according to the statistical characteristics of the RCS sequence, the scattering capability of the target to electromagnetic wave can be reflected, and is able to distinguish the target type.

As RCS modulation refers to the power modulation of LSS targets on radar echoes, we can use the value of SNR to represent the ability of power modulation of targets. Due to the influence of the distance between the target and radar, normalization of the value of SNR relative to range is necessary. Therefore, after obtaining the measured data of SNR and range from the actual radar system, we calculate the normalized SNR relative to range as the RCS value, shown as follows:(5)$$RCS=SNR+10\log R^{4}$$

Taking a helicopter, a fixed-wing UAV, a rotary-wing UAV, and a bird, for example, we calculate their respective RCS values in a period of time to obtain their RCS sequences based on the measured data. Then according to their respective RCS sequences, we calculate the statistical average of RCS sequences as the RCS feature, shown in Figure 9 as follows:

The distributions of RCS statistical averages for different types of targets have different ranges. From the measured data collected from the out-field, the RCS statistical average of helicopters is the largest, that of fixed-wing and rotary-wing UAVs is second, and that of birds is the smallest. We collect a set of measured data of helicopters, fixed-wing UAVs, rotary-wing UAVs, and birds, respectively, and calculate their statistical averages of RCS sequences, and then according to the averages, obtain their respective probability density curves. The probability density curves of statistical averages of RCS sequences of helicopters, fixed-wing UAVs, rotary-wing UAVs, and birds are shown in Figure 10 below:

It can be seen from the above result that the probability density curve distributions of the RCS averages of helicopters, fixed-wing and rotary-wing UAVs, and birds are in different ranges. Utilizing the statistical average of RCS sequences of a target in a period of time as its RCS feature can roughly classify the type of the target.

### 3.3. Motion Features

As holographic staring radar has the characteristics of continuous detection and high data rate, it can obtain more fine motion features of a target, so that it can extract the motion features of the target and classify the target type. The target parameters received by radar include range, velocity, azimuth angle, and elevation angle, so flying height, acceleration, and the change rate of acceleration (which is referred to as plus acceleration of the target) can be deduced. Since LSS targets belong to low altitude, small RCS, and slow speed targets, it is difficult to distinguish target types in terms of range, velocity, and height. Therefore, acceleration and relevant features can be used to characterize the mobility of target movement and assist in identifying target types.

For measured data, the number of intersection points between acceleration and plus acceleration sequence curves and zero is called the zero crossing number of acceleration and plus acceleration, which is used to express the drastic degree of acceleration and plus acceleration change. We set a capacity, count the number that the absolute values of acceleration and plus acceleration sequences that are greater than the capacity, which reflects the target’s mobility. It should be noted that the frame number of different targets collected in the experiment is different, which has a certain influence on the zero crossing number of acceleration and plus acceleration. Therefore, it is necessary to normalize the frame number of different target data, so we take the normalized zero crossing number of acceleration and plus acceleration on the frame number as the motion feature, representing the target’s mobility.

### 3.4. Multi-Dimensional Features

Multiple effective features are extracted from three dimensions of micro-Doppler, RCS, and motion to identify LSS targets. In the experiment, the SNR, range, velocity, acceleration, track position, and micro-Doppler spectrum of the targets can be obtained by the radar system at the output time of each datapoint. The above information can be divided into three categories: the RCS values reflect the electromagnetic scattering characteristics and posture characteristics of the targets; the micro-Doppler spectrum reflects more details of micro-motion parts of the targets; and the motion parameters reflect the motion characteristics and mobility of the targets.

A single RCS value of a target can reflect the scattering ability of the target to electromagnetic waves at the time of sampling. However, as RCS is greatly affected by the target’s posture angle, a single RCS value cannot distinguish different targets well. Therefore, the statistical average characteristic of RCS sequence can be selected as an RCS feature. The micro-Doppler spectrum can reflect the micro-motion of the target, and can reflect more detailed structure and motion information of the vibration or rotating parts on the target. Micro-Doppler features have a stronger identification ability. Acceleration and plus acceleration can roughly reflect the target’s mobility and motion range. Therefore, six multi-dimensional features are selected in this paper as the input features of the subsequent classifier for target recognition, as shown in Table 3:

## 4. Experimental Setup and Measured Data Processing

### 4.1. Experimental Setup

This experiment uses L-band holographic staring radar to collect measured data of LSS targets including helicopters, fixed-wing UAVs, rotary-wing UAVs, and birds. The adopted radar carrier frequency $f_{c}$ is 1.3 GHz, the working bandwidth is 2 MHz, and the pulse repetition frequency is 5 KHz. The measured data parameters include range, azimuth angle, elevation angle, velocity, RCS sequence, and micro-Doppler matrix data. We build a software operation platform to display the collected target data parameters, the playback of the target track, the Doppler view, etc. The interface of the software operation platform is shown in Figure 11 below:

### 4.2. Preprocess for Measured DMS

In the measured DMS, as the micro-Doppler frequency components are much weaker than the main frequency, it is difficult to observe directly the obvious micro-Doppler information in the raw Doppler data, so the raw Doppler data needs to be preprocessed, mainly including regularization and main frequency alignment for the DMS.

Taking a quadcopter for example, the raw measured DMS of the quadcopter is shown in Figure 12. This spectrum basically only displays the main frequency component generated by the fuselage movement, and it is difficult to observe the existence of micro-Doppler components generated by the rotary-wings, so we need to take the logarithm for raw measured DMS to obtain Figure 13. As can be seen from Figure 13, the logarithmic operation makes the weak micro-Doppler side frequency components more prominent, and more micro-Doppler details can be observed from DMS. However, though the amplitudes of the micro-Doppler components are increased, the amplitudes of the noise frequency components are also inevitably increased, making the weak micro-Doppler signatures difficult to highlight from the main frequency component and noises.

So we take a regularized log-Doppler spectrum [32]. We add a constant $C_{i}$ for the DMS, which is defined as taking the median of the DMS and then taking the logarithm. The micro-Doppler details are highlighted at the same time, suppressing the noise components. The constant $C_{i}$ and regularized log-Doppler spectrum are shown as follows, respectively, as well as the DMS in Figure 13 and Figure 14:
(6)$$C_{i}=median(S_{mD})$$
(7)$$S_{regularized\_mD}=\log\{S_{mD}+C_{i}\}$$

After regularization of DMS, the main frequency component and the micro-Doppler side frequency components can be clearly observed from the modulation spectrum in Figure 14. At this time, the noise frequency components are suppressed to $\log\{C_{i}\}$, and the micro-Doppler side frequency components have strong amplitudes, which can be highlighted from the noise, which is convenient for the subsequent extraction of micro-Doppler features.

The various velocities of the targets will be an obstacle for classification, so we need to take the main frequency alignment for the DMS. In other words, the main frequency component should be removed from the spectrum and the Doppler modulation spectrum is adjusted to align with respect to the zero frequency, shown in Figure 15.

From Figure 12 and Figure 15, after logarithmic processing, regularized logarithmic processing and main frequency alignment processing, the raw measured DMS, from which is difficult to observe obvious micro-Doppler information, is preprocessed into a DMS from which we can clearly observe the micro-Doppler side frequencies, so as to facilitate the extraction of micro-Doppler features in the next step.

After Doppler data preprocessing, according to Table 3, we extract multi-dimensional features from the collected and processed data. We make the maximum detection for the DMS after preprocessing to extract the number and interval of the maximum spectral lines, and the maximum micro-Doppler frequency. We calculate and extract the statistical average of the RCS sequence. We make a zero crossing detection for the acceleration and plus acceleration sequences, to extract the normalized zero crossing number of acceleration and plus acceleration on the frame number. Therefore, we can obtain six effective features extracted from three dimensions of RCS, micro-Doppler, and motion as the inputs of the classifier for target recognition.

### 4.3. Measured Data

The actual holographic staring radar system collects Doppler, range, velocity, SNR data, and other parameters of all targets in the whole observed airspace. The detection range can cover 20 km, the carrier frequency of radar is 1.3 GHz, its pulse repetition frequency PRF=5kHz, and the radar bandwidth is 2MHz. The measured Doppler waterfall plots, measured RCS sequence plots and measured acceleration and plus acceleration sequence plots from helicopters, rotary-wing UAVs and fixed-wing UAVs, and birds collected by the holographic staring radar system are shown in Figure 16, Figure 17 and Figure 18, respectively, as follows:

According to the above measured data, the ATR classification system can extract six effective features from the three dimensions of RCS, micro-Doppler, and motion automatically as inputs fed into a subsequent classifier system.

## 5. Multi-Layer Classifier System Design

The final step of the proposed ATR chain is to design an optimal classifier system and to perform classification with the multi-dimensional features mentioned above as the inputs fed into the classifier system.

Taking helicopters, fixed-wing UAVs, rotary-wing UAVs, and birds as examples, we design an optimal classifier system to realize the classification and identification of the four kinds of typical LSS targets. In the actual measurement of rotary-wing UAVs, some data contain obvious micro-Doppler information, while some data have no micro-Doppler information. For rotary-wing UAVs without obvious micro-Doppler information, their echoes are not significantly different from that of fixed-wing UAVs, both only including the main frequency component reflected by the fuselage. Therefore, rotary-wing UAVs are divided into two categories. The first category of rotary-wing UAVs represents the rotary-wing UAVs with micro-Doppler. The second category of rotary-wing UAVs represents the rotary-wing UAVs without micro-Doppler. Therefore, the collected measured target data are divided into five categories, as shown in Table 4.

### 5.1. Classification System Design

In terms of choosing the classifier, we choose the support vector machine (SVM), K-nearest neighbors (KNN), and back propagation (BP) neural networks as classifiers and we compare their respective performances in classification of the five LSS targets.

SVM uses the structure risk minimization criterion and the kernel trick to obtain a high accuracy of classification. However, SVM is a binary classifier, which cannot be generalized naturally to the multi-class classification case. There are two approaches which can be used to perform multi-class classification, i.e., one-versus-all SVM and one-versus-one SVM [33]. However, one-versus-all SVM can easily experience overlapping classifications or unclassifiable phenomenons. One-versus-one SVM has too many classifiers, and the number of classifiers is an order of magnitude $N^{2}$.Therefore, we innovatively propose a classification system based on multiple one-versus-one binary classifiers and a directed acyclic graph. The reasonable arrangement of the order of binary classifiers constitutes a fast and accurate classification system, as shown in Figure 19.

In the classification system for five types of LSS targets, an arbitrary two categories form a binary classifier, and the entire system has a total of ten binary classifiers. Then ten binary classifiers are arranged in a certain order to construct a directed acyclic graph classification system. The whole classification system is divided into four layers. The top layer is the root node, and each layer below has one more classifier than the upper layer, until finally the five categories of targets are all separated and the type of each target is identified. Each target only needs to pass through four classifiers, which can quickly complete classification and identification.

The disadvantage of this classification system is that if the classification result of root node classifier is wrong, it will cause error accumulation in each subsequent layer. In order to solve this problem, the selection of the root node classifier is particularly important. The binary classifier of the root node should select the two categories of targets with the largest differences, namely, the binary classifier with the highest classification accuracy. In this way, it is not easy to classify the root node incorrectly, and error accumulation can be greatly eliminated. Therefore, in this classification system, helicopters and birds, with the largest differences, should be selected as the binary classifiers of the root node, and then, from top to bottom of the classification system, we select in priority the binary classifiers with the largest differences of features and the highest classification accuracy, so as to constitute the whole classification system.

### 5.2. Classification Criterion

The first classification criterion for every binary classifier is based on multi-dimensional feature fusion classification result. The classification system in this criterion has ten binary classifiers, which have the same input features from one to six. We use the SVM classifier, KNN classifier, and BP neural network classifier and real measured data of five kinds of LSS targets to train and test the classification system. The classification accuracy rates of each binary classifier from one feature to six features are shown in Table A1 in Appendix A.

The experimental results in this table compare the classification accuracy of three different classifiers of SVM, KNN, and BP neural networks, and the classification accuracy corresponding to different feature numbers of the same classifier and the classification accuracy corresponding to two different types of targets. The following conclusions can be drawn after analysis:When the number of features is small, the SVM classification accuracy is higher than the KNN and BP network. When the number of features is large, the classification results of three classifiers are almost the same, as shown in Figure 20. In terms of time, KNN takes the shortest time, followed by SVM, and the BP network takes the longest time. Therefore, in the actual radar system, the SVM classifier is selected to ensure accuracy when there are less input features. When there are more features, the accuracies of three classifiers are similar and the KNN classifier with fast classification speed should be selected.For the six features selected by the system, the more the input features, the better the recognition effect of the classifier.The smaller the level of the classification system, the higher the classification accuracy, which indicates that the binary classifiers at the higher levels are more reliable and less prone to errors, thus reducing the probability of error accumulation in the classification system, shown in Figure 21. The experimental results verify the correctness and rationality of the sequential selection of the binary classifiers in the directed acyclic graph of the classification system.

The second classification criterion for every binary classifier is the equal-weighted comprehensive decision based on multiple single-dimension feature classification results. When every binary class classifier is in judging, we classify the target respectively from the three dimensions of RCS, micro-Doppler, and motion. Then according to the three respective classification results, we make a comprehensive decision with equal weights as the final classification result of the binary classifier. Then according to the classification system diagram in Figure 14, the multi-layer binary classifiers identify the type of the target successively until the last layer to identify the specific category of the target. The specific results are shown in Table A2 in Appendix B.

In Table A2, the experimental results describe the classification accuracy rates of each binary classifier in the three dimensions of RCS, micro-Doppler, and motion. In which, bird (B) vs. helicopter (H), B vs. rotary-wing UAV1 (R1), B vs. rotary-wing UAV2 (R2), and B vs. fixed-wing UAV (F), the four binary classifiers have high accuracies in RCS dimension. For B vs. H, B vs. R1, F vs. H, F vs. R1, R2 vs. H, R2 vs. R1, and R1 vs. H, the seven binary classifiers have high accuracies in the micro-Doppler dimension. For B vs. R1, B vs. R2, F vs. R1, and F vs. R2, the four binary classifiers have high accuracies in the motion dimension. Therefore, if we adjust the weights of different dimensions for different binary classifiers, the accuracy of the entire classification system will be improved.

### 5.3. Classification Results

We have collected 50 samples for each LSS target, respectively. To evaluate the classification performance of the proposed method, we use 60% of the total recordings as the training set and the remaining data as the validation set. This process is repeated 100 times. Each time we choose the train set and test set randomly. Then the average over these 100 repetitions is calculated to obtain the final classification accuracy rate. The classification accuracy rates using the SVM, KNN, and BP three different classifiers and two kinds of classification criteria are shown in Table 5, respectively.

The following conclusions can be drawn after analysis:For SVM, KNN, and BP, the multi-dimensional feature fusion classification system is better than that of the equal-weighted classification system with multiple single dimension feature classifiers, whether in classification time or accuracy.In the same classification criterion, KNN has a slightly faster classification time than SVM, and BP network classifier is the slowest, indicating that KNN and SVM are more suitable for classification systems with higher real-time requirements than the BP network.Comparing KNN and SVM, SVM has a higher classification accuracy than KNN in the equal-weighted classification system with multiple single-dimension feature classifiers, while SVM has a lower classification accuracy than KNN in the multi-dimensional features fusion classification system. It shows that SVM is better than KNN when there are less input features, and SVM is worse than KNN when there are more input features.

To sum up, when designing a radar ATR system for LSS targets, SVM and KNN classifiers are given priority to ensure the real-time performance of the radar system. Then, on the premise of not significantly increasing the classification time, more effective multi-dimensional features should be fed as far as possible as inputs into the classification system to improve the accuracy of classification. Based on the design principles summarized above, the classification system based on the multi-dimensional feature fusion of KNN classifier is the optimal classification system. Both the classification accuracy and the classification time are better than other design schemes, and the classification accuracy reaches 97.62%. Its confusion matrix is shown in Table 6.

## 6. Conclusions

A new automatic target recognition (ATR) system has been proposed for the classification of several LSS targets such as helicopters, fixed-wing and rotary-wing UAVs, as well as birds, based on the multi-dimension features of micro-Doppler, RCS, and motion. The influence of the number of rotary-wings on the amplitude and phase components of the radar echo was discussed and we concluded that the number of rotary-wings only affects the amplitude component of the radar echo signal, not the phase component in the far field. Then a complete ATR chain was proposed including the generation and preprocessing of DMS, multi-dimensional robust features extraction, a multi-layer classifier system design based on a directed acyclic graph; and finally, target classification. The results show that for the number of input features, the higher the number of input features, the better the classification result. For the KNN, SVM, and BP network classifiers, when the number of input features is relatively small, the SVM classifier is superior to the KNN and BP network classifiers. When the number of input features is large, the performances of the three classifiers are similar, but the KNN classifier has the shortest classification time. For feature dimensions, the features of different dimensions have different effects on different binary classifiers. For the two classification criterions, the multi-dimension feature fusion classification system is better than the equal-weighted classification system with multiple single-dimension feature classifiers, with higher classification accuracy and faster classification time. The final experimental results based on measured data show that the directed acyclic graph classification system based on the multi-dimensional feature fusion and KNN classifier is the best for LSS targets, in both classification time and accuracy, achieving the probability of correct classification on the order of 97.62%.

## Figures and Tables

**Figure 1 sensors-19-05048-f001:**
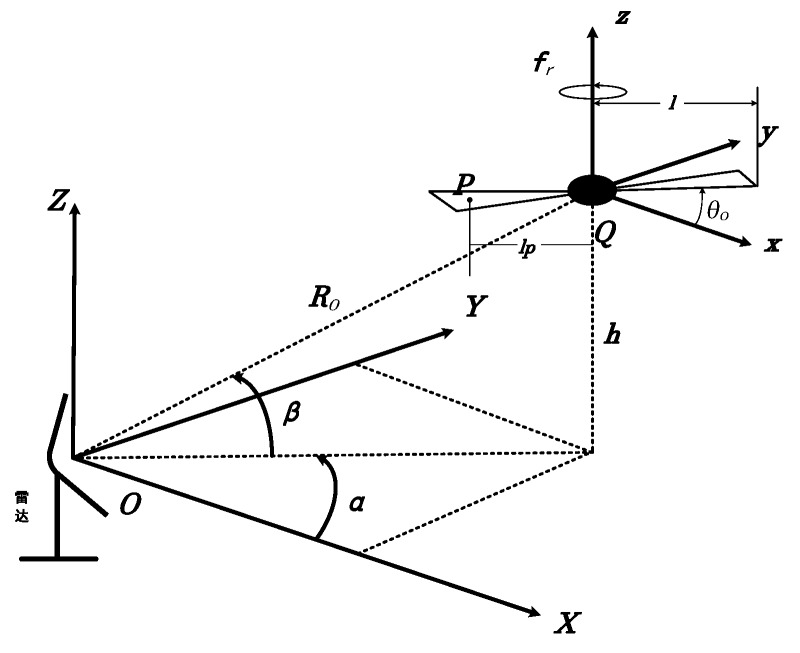
The geometry of the single rotary-wing and the radar.

**Figure 2 sensors-19-05048-f002:**
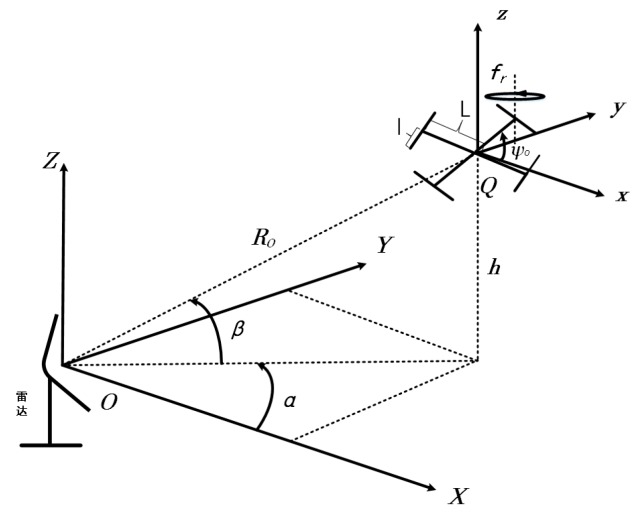
The geometry of the quad rotary-wings and the radar.

**Figure 3 sensors-19-05048-f003:**
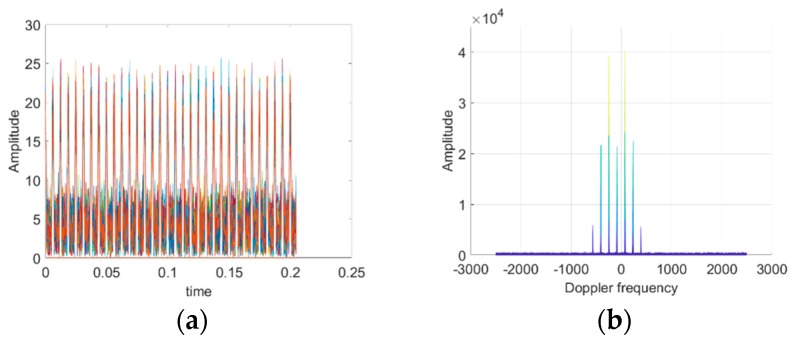
Simulation results of the ideal mathematical model of the single rotary-wing. (**a**) Time domain modulated echo. (**b**) Doppler domain modulation spectrum.

**Figure 4 sensors-19-05048-f004:**
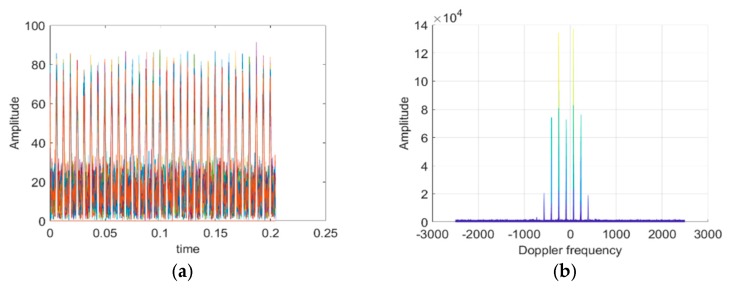
Simulation results of the ideal mathematical model of quad rotary-wings. (**a**) Time domain modulated echo. (**b**) Doppler domain modulation spectrum.

**Figure 5 sensors-19-05048-f005:**
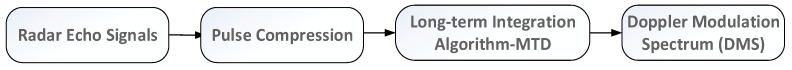
The specific process of obtaining the DMS.

**Figure 6 sensors-19-05048-f006:**
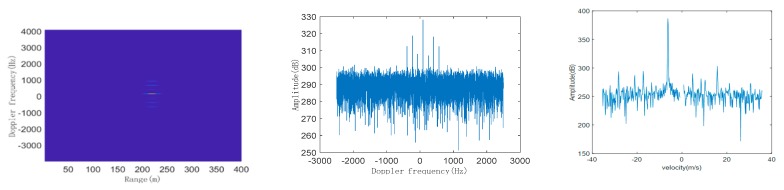
Simulated range-Doppler (RD) plane (**left**), simulated single-frame DMS (**middle**), and real measured DMS (**right**) of quadcopter.

**Figure 7 sensors-19-05048-f007:**
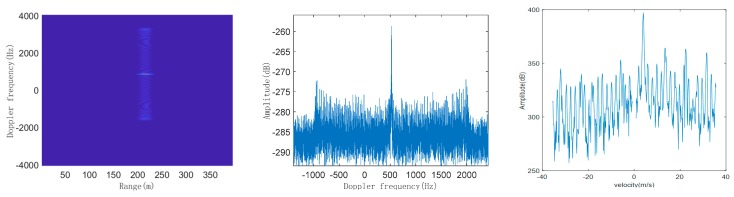
Simulated RD plane (**left**), simulated single-frame DMS (**middle**), and real measured DMS (**right**) of helicopter.

**Figure 8 sensors-19-05048-f008:**
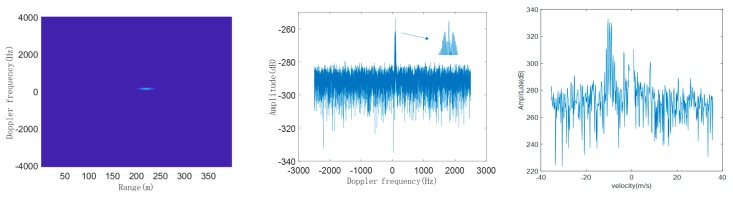
Simulated RD plane (**left**), simulated single-frame DMS (**middle**), and real measured DMS (**right**) of bird.

**Figure 9 sensors-19-05048-f009:**
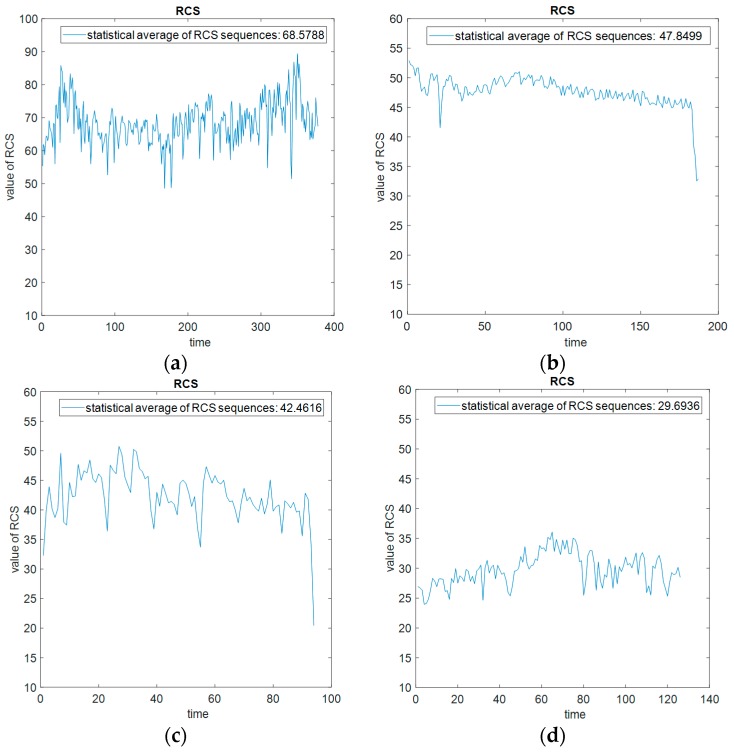
RCS sequences and respective statistical average of the measured data. (**a**) Helicopter. (**b**) Rotary-wing unmanned aerial vehicles (UAVs). (**c**) Fixed-wing UAV. (**d**) Bird.

**Figure 10 sensors-19-05048-f010:**
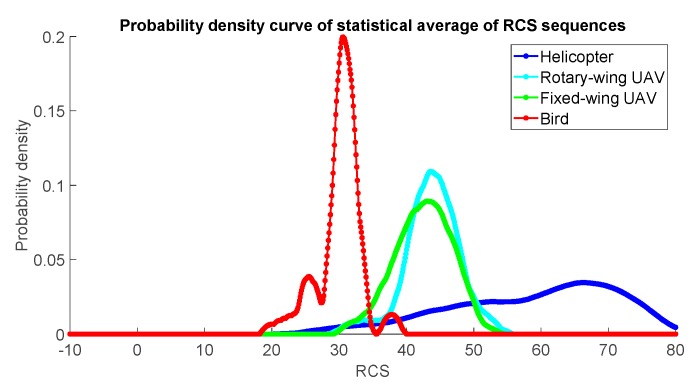
Probability density curves of statistical averages of RCS sequences of helicopters, fixed-wing and rotary-wing UAVs and birds based on measured data.

**Figure 11 sensors-19-05048-f011:**
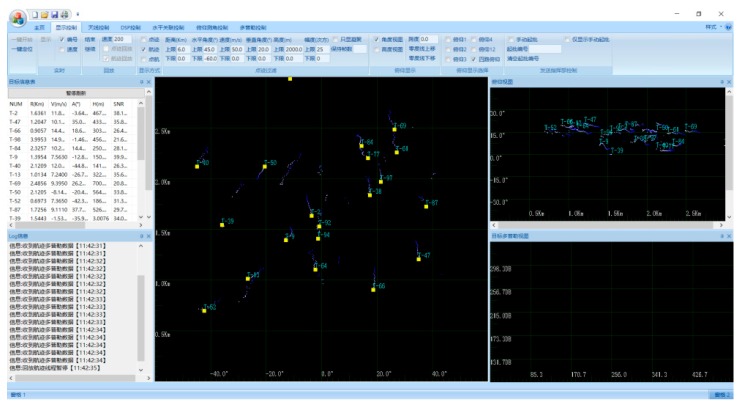
Interface of the software operation platform.

**Figure 12 sensors-19-05048-f012:**
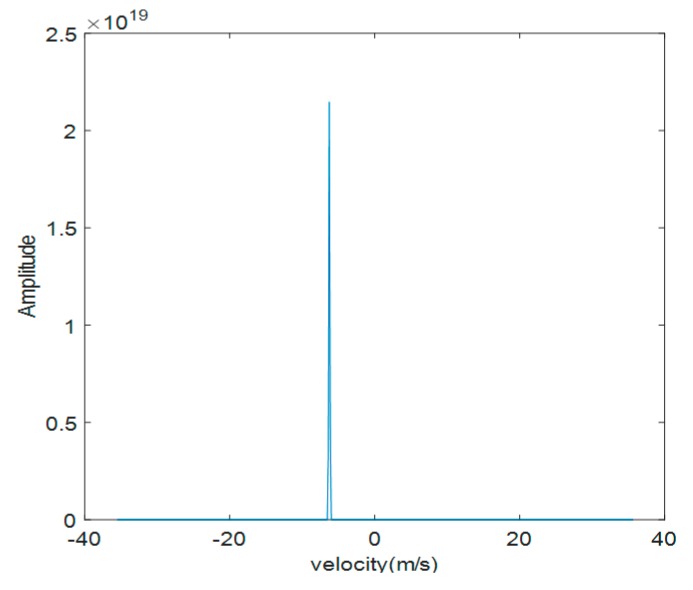
The measured raw DMS.

**Figure 13 sensors-19-05048-f013:**
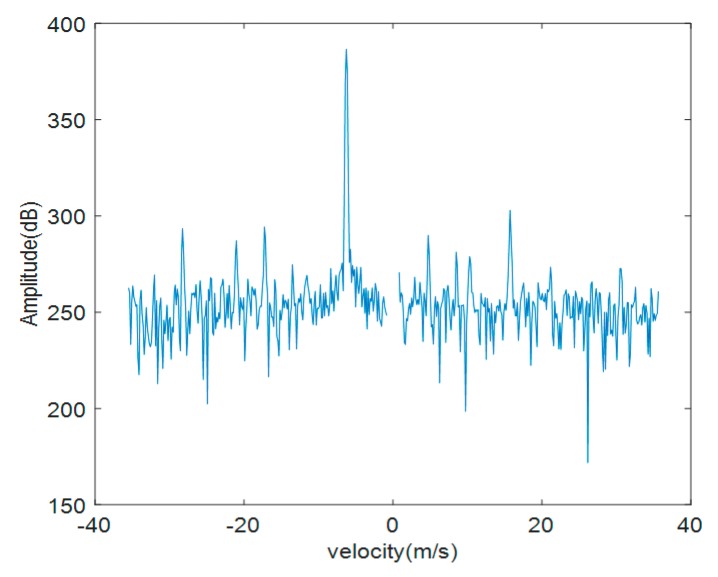
The logarithm for the measured DMS.

**Figure 14 sensors-19-05048-f014:**
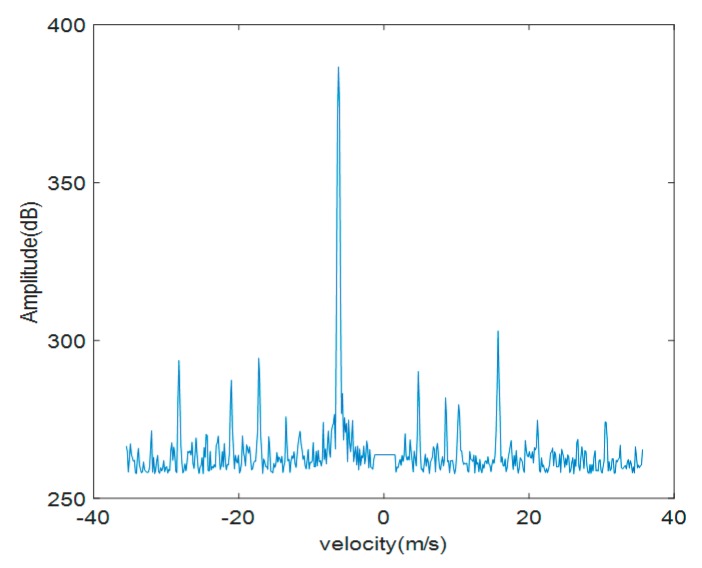
Regularized log-Doppler spectrum.

**Figure 15 sensors-19-05048-f015:**
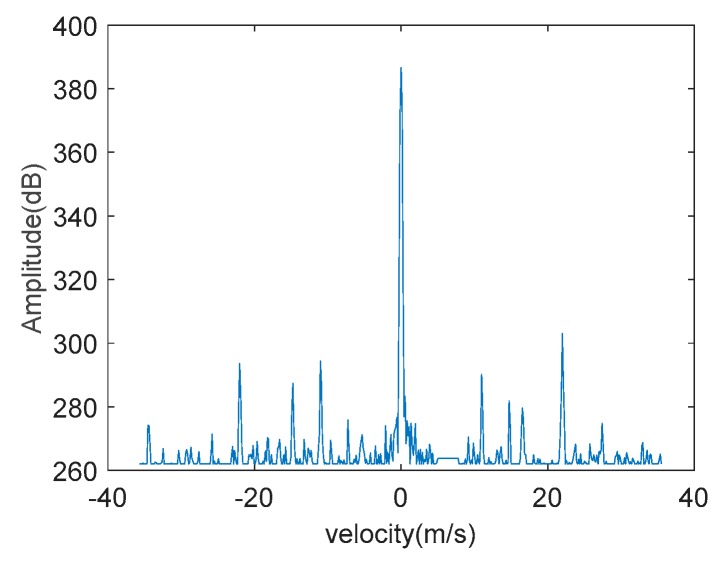
Regularization and main frequency alignment for DMS.

**Figure 16 sensors-19-05048-f016:**
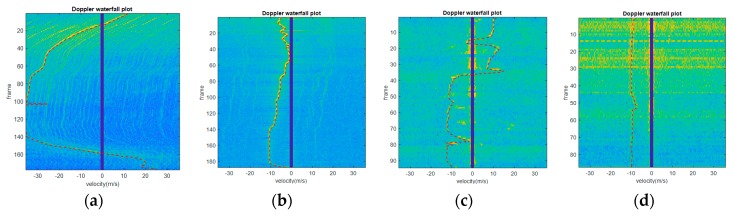
The measured Doppler waterfall plots. (**a**) Helicopter. (**b**) Rotary-wing UAV. (**c**) Fixed-wing UAV. (**d**) Bird.

**Figure 17 sensors-19-05048-f017:**
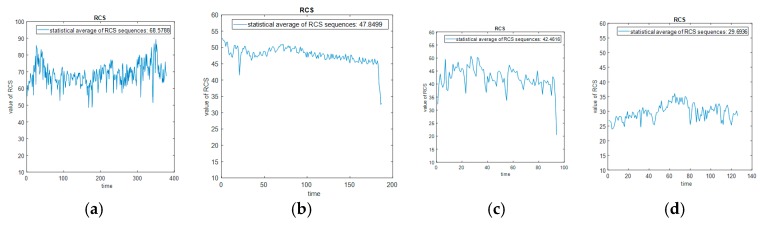
The measured RCS sequence plots. (**a**) Helicopter. (**b**) Rotary-wing UAV. (**c**) Fixed-wing UAV. (**d**) Bird.

**Figure 18 sensors-19-05048-f018:**
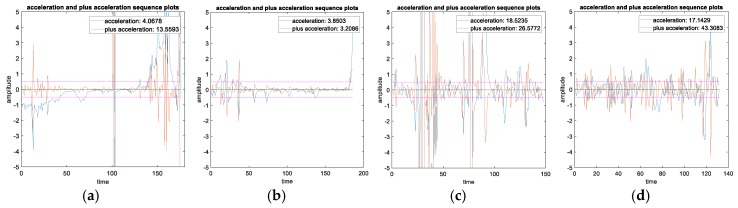
The measured acceleration and plus acceleration sequence plots. (**a**) Helicopter. (**b**) Rotary-wing UAV. (**c**) Fixed-wing UAV. (**d**) Bird.

**Figure 19 sensors-19-05048-f019:**
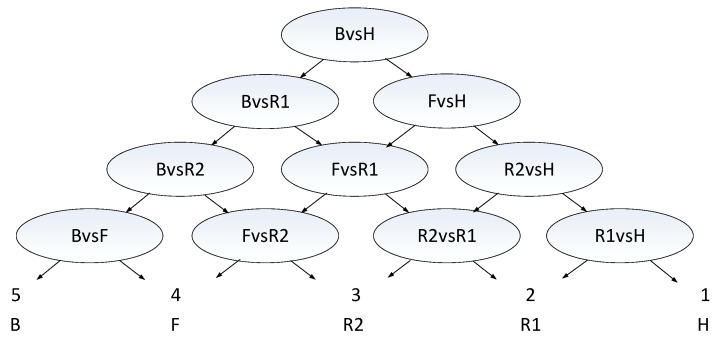
Classification system based on multiple one-versus-one binary classifiers and directed acyclic graph for five LSS targets.

**Figure 20 sensors-19-05048-f020:**
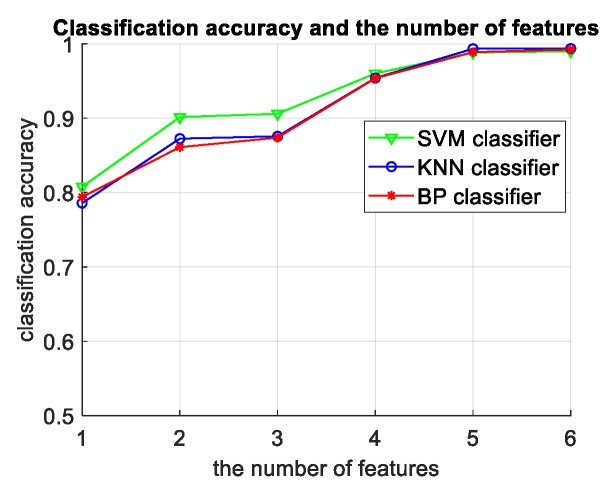
The relationship between classification accuracy and the number of features based on the measured data. Support vector machine (SVM), K-nearest neighbors (KNN), and back propagation (BP) neural networks.

**Figure 21 sensors-19-05048-f021:**
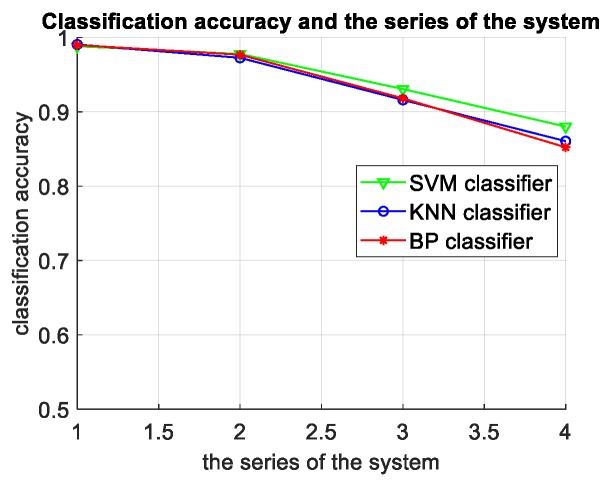
The relationship between classification accuracy and the series of the system based on the measured data.

**Table 1 sensors-19-05048-t001:** Radar and target parameters.

Radar Parameters		Parameter	Quadcopter	Helicopter	Bird
Radar frequency	1.3 GHz	Rotation rate	80 Hz	4.5 Hz	5 Hz
Radar bandwidth	2 MHz	Number of rotors	4	1	——
Pulse Repetition Frequency	5000 Hz	Number of blades	2	4	——
DotNum	400	Flight velocity	10 m/s	60 m/s	1 m/s
Number of pulses	8192	Blade length	0.1 m	6 m	0.5 m (upper arm)
					0.5 m (forearm)

**Table 2 sensors-19-05048-t002:** DMS feature comparison of three typical low altitude, small radar cross-section (RCS), and slow speed (LSS) targets.

Target Type	Spectral Line Number	Periodic Modulation Frequency	Spectral Width	Periodicity	Symmetry
Quadcopter	medium	large	wide	yes	yes
Helicopter	more	medium	extremely wide	yes	yes
Bird	less	small	narrow	no	yes

**Table 3 sensors-19-05048-t003:** Six multi-dimensional features for target recognition.

No.	Features	Dimension	Representation
1	RCS statistical average	RCS	Electromagnetic scattering ability
2	Spectral line number	Micro-Doppler	More detailed structure and motion information of micro-motion parts
3	Spectral width		
4	Periodic modulation frequency		
5	The normalized zero crossing number of acceleration on the frame number	Motion	Mobility
6	The normalized zero crossing number of plus acceleration on the frame number		Vibration

**Table 4 sensors-19-05048-t004:** The category of five LSS targets.

Category	Target	Abbreviation
Class 1	Helicopter	H
Class 2	Rotary-wing UAV1	R1
Class 3	Rotary-wing UAV2	R2
Class 4	Fixed-wing UAV	F
Class 5	Bird	B

**Table 5 sensors-19-05048-t005:** Classification accuracy and time of automatic target recognition (ATR) classification system with two classification criterions and the SVM, KNN, and BP neural network three classifiers.

Classification System	Equal-Weighted Classification System with Multiple Single-Dimension Feature Classifiers			Multi-Dimensional Features Fusion Classification System		
	SVM	KNN	BP	SVM	KNN	BP
Accuracy (%)	0.8116	0.7986	0.8342	0.9753	0.9762	0.9626
Time (s)	527.34	485.02	106,176.93	190.04	161.30	28,972.70

**Table 6 sensors-19-05048-t006:** Confusion matrix of five kinds of LSS targets in a multi-dimensional features fusion classification system based on the KNN classifier (%).

Target Type	H	R1	R2	F	B
H	100	0	0	0	0
R1	0	100	0	0	0
R2	0	0	95	5	0
F	0	0	5	90	5
B	0	0	0	0	100

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
