# Peer review of "A Complete Automatic Target Recognition System of Low Altitude, Small RCS and Slow Speed (LSS) Targets Based on Multi-Dimensional Feature Fusion"

_sensors, 2019, doi:10.3390/s19225048_

Round 1

Reviewer 1 Report

In this work, the authors propose a multi-classifier system based on the acyclic graph for small UAVs automatic classification. The developed idea is interesting, and the results obtained on real recorded data are encouraging, however the Reviewer has some concerns that need clarifications.

In the following the list of Reviewer's concerns and suggestions is reported:

The paper needs a careful read by the authors and an accurate proofreading to remove all typos and grammatical errors. Moreover, the English usage should be improved. The Reviewer suggests changing the acronyms sUAV in UAV, since most of the UAV are small targets. However, they could refer to them as small UAV and not sUAV. Please, do not start the sentences with “And”. The abstract is too long. Probably, it is better to shorten it. Moreover, it is not useful to give numerical results in the abstract such as “97.62%”. It is not appropriate to insert “LLS targets” in the list of keywords, since LLS is not a so famous acronym. Probably it is better to explicitly write it in that list. All the acronyms defined in the abstract should be also defined in the Introduction. On line 62 and 65, w-band must be W-band and x-band must be X-band (i.e., uppercase). The following related works on micro-Doppler based targets classification should be discussed for the sake of a complete review of the literature:

[1] “Analysis of radar micro-Doppler signatures from experimental helicopter and human data”, IET Radar, Sonar and Navigation, 1, 4, 289-299, 2007.

[2] “Micro-Doppler Effect in Radar”, Boston: Artech House, 2011.

[3] “GNSS based passive bistatic radar for micro-Doppler analysis of helicopter rotor blade”, IEEE Transactions on Aerospace and Electronic Systems, 50, 1, January 2014.

[4] “A novel algorithm for radar classification based on Doppler characteristics exploiting orthogonal pseudo-Zernike polynomials”, IEEE Transactions on Aerospace and Electronic Systems, 51, 1: 417-430, 2015.

[5] “On model, algorithms, and experiment for micro-Doppler-based recognition of ballistic targets”, IEEE Transactions on Aerospace and Electronic Systems, 53, 3, 1088-1108, 2017.

Please note that most of the symbols are not in math style, e.g., N in line 112, Q and L in line 155, and so on. Please check the entire paper. The first part of Section II could be simplified by reducing the formulas (1)-(3). Since they are already known and reported in several papers and books (as those highlighted above), I think that it is sufficient to directly write equation (3) and properly citing some references. Note that the imaginary unit is sometimes indicated with the letter “i” and sometimes with letter “j”. I suggest using always the same symbol for reader’s easy. Just after equation (5), the delta distribution should be defined. On line 162 some badly written symbols are present. Please check it. The paragraph from line 169 to 192 probably is unnecessary. Therefore, it could be removed or moved to the analyses section. Please, add the units of measure on the axes of figures 6-8. In which way the RCS is computed? Please, add some details.

Reviewer 2 Report

Detailed Comment;

References 18-21 (referred at line 103) are not for helicopter signatures – please add references at helicopter in that point rather to body vibration

The considerations presents in eq (1) to (7) are well known – ant in the paper it seems to be own achievement of the author. As the rotor are continues, it should be form as the integral rather than a sum

Please refer to previous publications and/or explain differences (if any)  

Line 172 – “and sampling interval is 0.2ms” it is not precise, please explain (with 2 MHz bandwidth sampling interval should be much higher).

The simple simulations (using eq-17) ware presented almost 3 decades ago) more valuable would be result of full electromagnetic simulations.

Line 280 – what is the meaning of “statistical average of RCS sequences”  

Line 328 ??? “1.3ghz”

Fig 10 – amplitude (in dB) 2.5*10^19 – is not realistic,

In the paper there is no info if it is measured or simulated.

Also on other plots  (11-13) 150 dB is questionable – please explain

It would be nice also to see measured signals from birds, helicopter, rotary wing and fixed wings.

It is not clearly stated if the results are based on measurements (an with which conditions) or simulations.

Round 2

Reviewer 1 Report

In this work, the authors propose a multi-classifier system based on the acyclic graph for small UAVs automatic classification. The developed idea is interesting, and the results obtained on real recorded data are encouraging. The authors have modified the paper according to the Reviewers’ suggestions; however, some minor comments remain, that I've listed in the following:

Some typos are still present in this version of the paper. For instance, “micromotions” on line 58 should be “micro-motions” according to the previous definition. On line 65 probably “it purposes” could be “it proposes”. However, I suggest another careful reading of the paper by the authors or, if possible, by a native English in order to remove all typos and grammatical errors. Is it possible to increase the size of the subplots in Fig. 9? They are difficult to read. As to the last references that you have added. Please, can you write all the authors’ name? Some names are missed.

Author Response

Point 1: Some typos are still present in this version of the paper. For instance, “micromotions” on line 58 should be “micro-motions” according to the previous definition. On line 65 probably “it purposes” could be “it proposes”.

Response 1: I am so sorry that I have so many typos in my paper and the typos cause you some trouble in reading. I have tried my best to correct the typos and grammatical errors in the paper. Besides, I have asked the authors of the paper to read carefully and correct the typos in the entire paper.

Point 2: Is it possible to increase the size of the subplots in Fig. 9?

Response 2: In the previous paper, I set Fig. 9 relatively small, in order to save space. I am so sorry to cause you some trouble in reading. I have increased the size of the subplots in Fig. 9 in new paper.

Point 3: As to the last references that you have added. Please, can you write all the authors’ name? Some names are missed.

Response 3: I have added all the authors’ name of the last references that I added.

Reviewer 2 Report

References 18-21 (referred at line 103) are not for helicopter signatures – please add
references at helicopter in that point rather to body vibration. references [1-4] are relatively new - it would be nice to have reference to the first works not only to the followers !!!!

Fig 12 (prev 10) – amplitude (in dB) 2.5*10^19 – is not realistic, explanation is OK but no changes in the paper !!!!!!

Table 1 - units are not consistent

Rotation rate 80rad/s 4.5rad/s 5Hz

Round 3

Reviewer 2 Report

Caption in Fig 1 is "amplitude (dB) and value is 2 10^19

This value is not valid - please correct
